# Psychological profiles of South African smallholder farmers

**Navjot Bhullar**[1,2]*, **Nkhanedzeni B. Nengovhela**[3,4], **Livhuwani Mudau**[3], **Renato A. Villano**[5], **Isaac Koomson**[5], **Heather M. Burrow**[5]

**1** School of Psychology, University of New England, Armidale, Australia, **2** Discipline of Psychology, Edith Cowan University, Perth, Australia, **3** Department of Agriculture, Land Reform and Rural Development, Riviera, South Africa, **4** School of Agriculture & Life Sciences, University of South Africa, Pretoria, South Africa, **5** UNE Business School, University of New England, Armidale, Australia

* n.bhullar@ecu.edu.au

**Data Availability Statement:** Anonymized data are publicly available on the figshare database (https://doi.org/10.6084/m9.figshare.19354313).

**Funding:** Initials of the authors who received each award: HMB for the ACIAR project and NBN for the

## Abstract

The present study examined smallholder farmer profiles based on key psychological variables associated with farm business performance in the South African context. A sample of 471 beef farmers (mean age = 54.15 years; *SD* = 14.46; men = 76%) and 426 poultry farmers (mean age = 47.28 years; *SD* = 13.53; women = 54.5%) provided data on a range of measures assessing attitudes, subjective norms, perceived behavioral control, personality characteristics, present and future time orientation, expected benefits of, and efficacy to perform the farm business tasks, and farm-related concerns. Latent profile analysis identified three distinct profile segments of smallholder beef and poultry farmers, respectively: *Fatalists*, *Traditionalists*, and *Entrepreneurs*. Our results suggested unique combinations of psychological characteristics in a sample of South African smallholder beef and poultry farmers and show a novel way of understanding enablers of, and barriers to, engaging in the farm business.

## Introduction

South Africa's beef and poultry industries are segregated into three 'economies': (i) a commercial sector with well-developed value chains, industry infrastructure and production systems equal to those of most developed countries; (ii) a 'commercially-oriented' smallholder sector comprising farmers who own or lease land and are commercially-oriented but lack the training, infrastructure and production systems available to the commercial sector; and (iii) a 'communal farmer' sector, where smallholder farmers use communally-owned land for livestock production and operate mainly as subsistence farmers.

In 2016, there were 13.6 million cattle in South Africa, with 42% (5.7 million animals) owned by smallholder farmers, who marketed less than 10% of their animals each year, compared to 25% in the commercial sector [1]. These figures demonstrate strong potential for improved beef production to benefit the farmers, their communities, beef supply chains and ultimately, South Africa's economy.

DALRRD funds Grant numbers awarded to each author: ACIAR LPS/2016/276 and no grant number for the DALRRD funds The full name of each funder: Australian Centre for International Agricultural Research; South African Department of Agriculture, Land Reform and Rural Development URL of each funder website: https://www.aciar.gov.au/ and http://www.dalrrd.gov.za/ (ARC is the administrator of DALRRD funds for the poultry recording scheme) Did the sponsors or funders play any role in the study design, data collection and analysis, decision to publish, or preparation of the manuscript? The funders had no role in study design, data collection and analysis, decision to publish, or preparation of the manuscript.

**Competing interests:** The authors have declared that no competing interests exist.

South Africa's poultry industry is the largest single contributor to the country's agricultural sector. In 2019, 20% of South Africa's total agricultural gross value and 41% of animal product gross value stemmed from poultry production [2]. About 75% of birds in the South African poultry industry were used for meat production, while the remaining 25% were used for egg production. In sharp contrast to the beef industry, smallholder farmers produce only 2% of South Africa's chicken meat and less than 0.5% of South Africa's egg production [2]. However, broiler and egg production offer many new opportunities for South Africa's poor populations living in rural communities and experiencing high unemployment rates because start-up costs for the poultry industry are substantially less than those in the beef and other agricultural sectors.

Development of smallholder farmers is, therefore, a very high priority for the South African government. Since the end of apartheid in South Africa in 1994, successive South African governments have invested strongly in smallholder farmer support schemes, including land, finance, and infrastructure costs as well as information and marketing infrastructure. However, livestock production amongst smallholder farmers remains very low.

In an attempt to identify reasons for the persistent low levels of productivity amongst smallholder farmers in South Africa, Nengovhela [3] investigated the barriers and promoters of individual change considered essential to improve the wellbeing of individual smallholder beef farmers in one South African province (Limpopo). Focus group discussions and in-depth interviews were undertaken amongst different types of commercial and smallholder farmers as well as research and extension staff tasked with supporting those farmers. Results from that study showed there was a need to build human, social, financial and psychological capital amongst smallholder farmers. Critically, when farmers had equivalent access to land, water, infrastructure, knowledge and finance, psychological capital had the greatest impact on adoption of practices needed to improve the business performance of smallholder beef farmers [3]. That study recommended smallholder farmers be provided with customized coaching to assist them to improve their farm business performance.

Due to the very significant challenges of implementing customized coaching to millions of smallholder farmers, the present study instead aimed to determine the value of using psychological profiles of individual farmers, using the methods outlined by Morgan and colleagues [4] to segregate smallholder beef and poultry farmers in South Africa into groups of farmers with similar behavioral profiles and then to determine whether relationships existed between the farmers' behavioral profiles and their farms' business performance. The current study primarily focuses on the development of psychological profiles for commercially-oriented smallholder beef and poultry farmers in South Africa.

## Key psychological predictors of farm business performance

To explore the predictors of business performance, an understanding of psychological models of human behavior is required. Existing studies have shown that psychological characteristics of farmers are significantly associated with adoption of best agricultural practices [5–7] and eventual performance of their farm businesses [7–9].

Regarding farmers' adoption of best practices, a study of sheep farmers in Scotland by Jack and colleagues [5] examined how psychological factors influence adoption of best farming practices and found that farmers' perceived expectation (social norms) of adoption by neighboring farmers had the greatest relative effect on their adoption of parasite control practices such as quarantine strategy for parasite control and the use of parasite diagnostic testing for monitoring faecal egg counts and detecting anthelmintic resistance. Another study employed the Extended Theory of Planned Behavior Model on crop farmers in Bangladesh and found

that attitude and perceived climatic threats of conventional farming influence farmers intention to adopt conservative agricultural practices [6].

Considering farm business performance, O'Leary and colleagues [8] investigated the link been three personality measures and farm financial performance and found that conscientiousness, leadership, and relaxed personality traits are associated with increases in farm profit for dairy farmers in England and Wales. In a related study, Suksod and colleagues [9] found that psychological capital (i.e., hope, resilience, confidence, and optimism) serves as an important pathway through which agricultural extension knowledge influences farm performance among crop farmers in Thailand. A further study of wheat farmers in Australia also showed that socio-psychological variables do not only cause yield gaps directly but also influence farm management practices, which in turn cause yield gaps [7].

Our study specifically focuses on four main psychological theories relevant to agriculture or farm business performance. These are: (i) Theory of Planned Behavior [10,11] (TPB) comprising three psychological (cognitive) predictors of behavioral intentions: attitudes (an evaluative disposition towards the target behavior), subjective norms (beliefs as to whether significant others (e.g., family and friends) approve (or disapprove) of them undertaking the behavior) and perceived behavior control (the extent to which individuals believe that they have control over the target behavior). TPB has been successfully applied to explain a wide range of behavioral intentions, including conservation farming [12]; (ii) Five-Factor Personality Model [13,14] assesses personality traits that underscore an individual's characteristic way of thinking, feeling and behaving [15]. It encompasses five personality characteristics: Openness to experience (engaging in the search and pleasure of new experiences), Conscientiousness (abilities associated with organization, perseverance, impulse control, and respect for environmental standards), Extraversion (quantity and intensity of relationships with one's environment and the level enthusiasm and energy displayed during environmental interaction), Agreeableness (quality of one's interpersonal relationships from compassionate to antagonistic), and Neuroticism (susceptibility to construct and perceive one's environment as harmful, difficult, and threatening); (iii) Time orientation [16] reflects an attention bias towards past, present or future outcomes through which an individual makes meaning from experiences and external events. Individuals tend to be strongly oriented towards a single temporal framework [17], with present-time orientation (tendency for immediate gratification and pessimistic future outlook) and future-time orientation (tendency to delay immediate gratification for long-term benefits) found to be related with agricultural practices [4]; and (iv) Self-efficacy [18,19] describes an individual's belief about their ability to engage in a particular behavior. Self-efficacy has been found to be linked with a range of agricultural behaviors, e.g., agroforestry adoption and maintenance [20] and low emission agriculural practices [4].

In addition, we included expected benefit considerations for engaging in the farming business. Evidence suggests that positive expectancies are important determinants of behavior, including financial pay-offs, maximizing income and minimizing costs as indicators of success of farming operations [4]. We also investigated the role of farming concerns related to personal capability and those specifically related to the farm business [21].

## The current study

It is evident that single individual-level characteristics such as attitudes, personality traits and time orientation have been examined in earlier research in relation to farm business performance. To our knowledge, no empirical research in this area has examined profile-based combinations of a number of psychological characteristics that combine or co-exist within a person. That requires an exploration of psychological profiles. This novel approach is taking

the field beyond bivariate associations, and helping to identify farmer typologies that can provide new insights into how different psychological characteristics combine or co-exist within an individual.

Traditional variable-centered approaches explore the relationships between variables, thus, limiting our ability to form inferences about individuals [22,23]. For instance, a regression-based approach examines the main effects and interactions. However, it does not imply that the "groups" (with high scores on one variable and low on another) obtained in a moderation analysis are always meaningful. On the other hand, person-centered approaches, such as a *latent profile analysis* (LPA) identifies specific combinations of variable scores that occur naturally within a sample and group respondents with similar scores across a set of variables. LPA classifies individuals into homogenous, probability-based groupings and examines the relationships between individuals and their different patterns of responses [24]. LPA provides a novel approach to examine the prevalence of different patterns of responses on a range of individual difference variables [25,26].

## Method

### Participants and procedure

Our sample comprised smallholder 471 beef (mean age = 54.15 years; *SD* = 14.46; men = 76%) and 426 poultry (mean age = 47.28 years; *SD* = 13.53; women = 54.5%) farmers in South Africa. This study is part of a large Australian Centre for International Agricultural Research (ACIAR)-funded project to improve the profitability of emerging and smallholder cattle beef farmers by developing cost-effective and environmentally-sustainable beef value chains that supply cattle to meet the specifications of high-value, free-range beef markets. The poultry farmers were supported by a South African government-funded project to support farming through a Poultry Improvement Scheme. The main objective of the scheme is to expose and equip egg and broiler producers with the latest technology and science in farming, thus enabling egg and broiler producers to operate efficiently and profitably. This scheme is established to ensure and encourage data collection and recording at farm level that should help measure and compare individual enterprise performance.

The beef farmers were located within a 250-km radius of Cradock Abattoir (Eastern Cape Province) and the similar radius of Cavalier Meats (which sources supply from Gauteng, Mpumalanga, Limpopo, North West and Free State Provinces), with both abattoirs supplying cattle into the free-range markets promoted by Woolworths to middle-high income consumers. Poultry farmers were from the same provinces as the beef farmers. Most of the respondents have attained at least secondary education (beef = 75%, poultry = 73%) and an average of 17.5 years and 4.25 years in beef and poultry farming, respectively. The ethnic composition and language background of respondents were almost the same for beef and poultry farmers, where Sepedi, Setswana, Isizulu, IsiXhosa and Sesotho dominates the households' ethnic composition.

Ethics approval to use the data was obtained from the University of New England Human Research Ethics Committee (HREC Approval Number: HE19-081 valid to 09 May 2022). The study survey was prepared in the English language. For data collection, the survey questions were verbally translated by the trained enumerators from the project sites and communicated to participants in their local language if the participants (farmers) were not confident of their English language skills—this was applicable to only those participants who had limited schooling. Most of the people in South Africa have been educated through English speaking schools and are fluent in English language.

## Measures

In addition to demographic variables, the following measures were used. Internal reliability of the measures used in the present study was assessed by Cronbach's $\alpha$s and are presented in Table 1.

**Attitudes.** Attitudes were assessed by 5 items measured on a 5-point Likert scale (1 = strongly disagree; 5 = strongly agree) and averaged to compute an overall score, with high scores indicating positive endorsement of attitudes towards beef/poultry farming. Sample items are "*I feel a strong responsibility to continue in beef/poultry farming*", "*I feel morally obliged to continue in beef/poultry farming*". Previous research [27] has suggested excellent internal reliability (Cronbach's $\alpha$ = .91).

**Subjective norms.** Subjective norms were assessed by three items measured on a 5-point Likert scale (1 = strongly disagree; 5 = strongly agree) and are averaged to compute an overall score, with high scores indicating positive endorsement of subjective norms towards beef/poultry farming. Sample items are "*Most people who are important to me think I should continue in beef/poultry farming*" and "*Most people who are important to me would support my participation in beef/poultry farming*". Previous research [28] has suggested satisfactory internal reliability (Cronbach's $\alpha$ = .77).

**Perceived behavioral control.** Perceived behavioral control was assessed using 2 items measured on a 5-point Likert scale (1 = strongly disagree; 5 = strongly agree) and are averaged to compute an overall score, with high scores indicating positive endorsement of perceived behavioral control towards beef/poultry farming. The two items were "*I feel I have control over whether or not I continue in beef/poultry farming*" and "*I am confident I can perform activities*

**Table 1. Intercorrelations, means, Standard Deviations (SD) and Cronbach's $\alpha$s of key study variables for beef farmers.**

| Variables | 1. | 2. | 3. | 4. | 5. | 6. | 7. | 8. | 9. | 10. | 11. | 12. | 13. | 14. |
|---|---|---|---|---|---|---|---|---|---|---|---|---|---|---|
| 1. Attitudes | - | .59*** | .60*** | .23*** | .28*** | .20*** | .01 | .25*** | .16*** | .50*** | .40*** | .58*** | .28*** | .30*** |
| 2. Subjective norms | | - | .73*** | .26*** | .25*** | .08 | .04 | .03 | -.01 | .51*** | .37*** | .57*** | .28*** | .28*** |
| 3. Perceived behavioral control | | | - | .31*** | .33*** | .11* | .07 | .02 | -.12** | .59*** | .41*** | .57*** | .24*** | .25*** |
| 4. Openness[+] | | | | - | .32*** | .16** | .18*** | .23*** | .01 | .39*** | .13** | .22*** | .02 | .01 |
| 5. Conscientiousness[+] | | | | | - | .15** | .17*** | .07 | -.02 | .34*** | .27*** | .24*** | .07 | .01 |
| 6. Extraversion[+] | | | | | | - | .17*** | .15** | .25*** | .08 | .07 | .14** | .16** | .13** |
| 7. Agreeableness[+] | | | | | | | - | -.10* | .03 | .09 | .04 | .09 | .06 | .08 |
| 8. Neuroticism[+] | | | | | | | | - | .29*** | -.01 | .03 | .14** | .15** | .12** |
| 9. Present time | | | | | | | | | - | -.08 | -.02 | .09 | .24*** | .21*** |
| 10. Future time | | | | | | | | | | - | .42*** | .47*** | .22*** | .25*** |
| 11. Expected benefits | | | | | | | | | | | - | .48*** | .33*** | .28*** |
| 12. Self-efficacy | | | | | | | | | | | | - | .39*** | .32*** |
| 13. Personal capability concerns | | | | | | | | | | | | | - | .67*** |
| 14. Farm-related concern | | | | | | | | | | | | | | - |
| **Mean** | 3.79 | 4.08 | 4.08 | 4.04 | 4.17 | 3.37 | 3.77 | 3.18 | 2.84 | 3.74 | 3.86 | 3.74 | 3.06 | 3.22 |
| **SD** | .69 | .74 | .77 | .98 | .96 | 1.20 | 1.17 | 1.22 | .77 | .50 | .79 | .64 | .80 | .76 |
| **Cronbach's $\alpha$** | .70 | .80 | .73 | - | - | - | - | - | .74 | .73 | .87 | .86 | .90 | .89 |

Note.

*$p < .05$,

**$p < .01$,

***$p < .001$.

[+] 1-item measure, therefore, no Cronbach's $\alpha$ was computed.

*related to beef/poultry farming if I wanted to*". Previous studies [27,28] have reported Cronbach's αs ranging from .77 to .88.

**Personality.** Personality traits were assessed using a brief version of the Big Five Personality Inventory-10 (BFI-10) [29], namely Openness to experience, Conscientiousness, Extraversion, Agreeableness, and Neuroticism (OCEAN). Each personality trait was assessed by 2 items. Items, in each subscale, are measured on a 5-point Likert scale (1 = disagree strongly; 5 = agree strongly) and averaged to compute an overall score, with high scores indicating endorsement of a specific personality characteristic. Examples of sample items are "*I see myself as someone who is outgoing, sociable*" (Extraversion) and "*I see myself as someone who does a thorough job*" (Conscientiousness). Previous research [30] examined the association between the five personality traits and entrepreneurial success, operationalized as business growth, profitability and financial performance, in South Africa, and found Cronbach's αs ranging from .53 to .71. In our study, we found the smallholder farmer respondents were not able to effectively interpret and understand all negatively-worded items as the Cronbach's αs obtained for those items were close to 0. Therefore, we decided to use only the positively-worded 1-item for each of the Big 5 traits in the present study and no Cronbach's α was computed for 1-item positively-worded statements of the Big Five.

**Time perspective.** Time Perspective was measured using two subscales from the Zimbardo Time Perspective Inventory [16]: present- and future-time orientation. Present time orientation was assessed by 8-items. Sample items include "*I take risks to put excitement into my life*" and "*I do things impulsively, making decisions on the spur of the moment*". Future-time orientation was assessed by 13-items. Sample items include "*I am able to resist temptation when I know there is work to be done*" and "*I complete projects on time by making steady progress*". Items are assessed on a 5-point Likert scale ranging from 1 = Untrue to 5 = Very true, and are averaged to compute a composite score for each of the subscale, with high scores indicating a tendency to live in the present time or future time, respectively. Previous research [31] demonstrated a good internal reliability (Cronbach's α = .80 for present time and α = .75 for future time).

**Self-efficacy.** Self-efficacy was assessed by an 11-item measure adapted from New General Self-Efficacy Scale [4,32]. Items are measured on a 5-point Likert scale (1 = strongly disagree; 5 = strongly disagree) and averaged to compute an overall score, with high scores indicating self-efficacy to execute a plan of action in the farming business. Sample items include "*I will be able to achieve most goals that I have set for myself*" and "*When facing difficult tasks, I am certain that I will accomplish them*". Previous research[4] showed excellent internal reliability (Cronbach's α = .91).

**Expected benefits.** Expected benefits of engaging in beef/poultry farming were assessed using an 8-item measure capturing financial benefit (adapted from Morgan and colleagues [4]) and benefits to the community and environment—items developed specifically for the present study. Items are measured on a 5-point Likert scale (1 = very unlikely; 5 = very likely) and averaged for a compositive measure with high scores indicating perceptions of positive benefits of engaging in beef/poultry farming. Sample items include: "*I will be able to have a reliable source of income*" and "*It will be beneficial to my community/ the environment*". Previous research [4] found excellent internal reliability for the financial benefit scale (Cronbach's α = .89).

**Farming concerns.** Farming concerns were assessed using a 29-item measure adapted from Morgan and colleagues' study [4] examining farmer stress dimensions [21]. In the present study, two subscales were used: *personal capability concerns* (16 items) and *farm-related concerns* (13 items). Items are measured on a 5-point Likert scale (1 = very low; 5 = very high) and averaged to compute a scale score for each of the subscales with high scores indicating

high personal capability and farm-related concerns. Sample items include: "*Personal illness during peak times*" (Personal Capability concern) and "*Availability of reliable and skilled farm labour*" (Farm-related concern). Previous research[21] found excellent internal reliabilities for the farmer stress dimensions (Cronbach's αs ranging from .79–.91).

## Statistical analyses

First, we conducted descriptive statistics (Means and *SD*) and bivariate correlations (Pearson's *r*) among key study variables for beef and poultry farmers. Second, we used a latent profile analysis (LPA) using Mplus [33] (v.7.3) to classify respondents based on shared patterns of their responses on a range of psychological characteristics. LPA is a sophisticated analytical tool used to assess how unique combinations of continuous latent variables and underlying categorical latent variables cluster within homogeneous groupings within a sample. Several model fit indices were assessed to determine the optimal profile model, including the Bayesian Information Criteria (BIC), which assesses improvement in fit after adjusting for the number of parameters in a model, sample size adjusted BIC [34,35], Vuong-Lo-Mendel-Rubin (VLMR) Adjusted test, and the Bootstrapped Likelihood Ratio test (BLRT). The VLMR and BLRT assess differences in goodness-of-fit between model *k* and model *k*-1, where *k* refers to the number of retained profiles. The preferred model is indicated by a combination of smallest BIC and adjusted BIC values with highest number of profiles. Significant *p* values for LMR and BLRT indicate best fit, i.e., model *k*-1 should be rejected in favor of model *k* [33]. Entropy was also used as an index of model assessment, with values close to one considered ideal [36]. In addition to statistical adequacy, we also considered theoretical conformity and meaningfulness and interpretability of the preferred profile-solution to guide our decision regarding retaining the number of profiles [37–39]. To facilitate interpretation of profiles, we standardized all 14 psychological profiling variables to a mean of 0 with a standard deviation of 1.

Finally, two multivariate analyses of variance (MANOVAs) were conducted to examine significant differences in the profiling variables across the three profiles of beef and poultry farmers, respectively.

## Results

### Descriptive statistics

Table 1 shows intercorrelations, means, and standard deviations of key study variables for beef farmers. All correlations were in the expected direction. Specifically, individuals who were future-oriented reported positive attitudes, subjective norms, perceived behavioral control, positive minded (openness to experiences and conscientiousness), and expected benefits of and self-efficacy in engaging in farming. However, there were no significant associations between attitudes and agreeableness; between subjective norms and extraversion, agreeableness, neuroticism and present time orientation; between perceived behavioral control and agreeableness and neuroticism; between openness to experience and present time orientation, personal capability and farm-related concerns; between conscientiousness and neuroticism, present-time orientation, personal capability and farm-related concerns; between extraversion and future time orientation and expected benefits; between agreeableness and present- and future-time orientation, expected benefits and self-efficacy, personal capability and farm-related concerns; between neuroticism and future time orientation and expected benefits; and between present time orientation and future time orientation, expected benefits and self-efficacy.

Table 2 provides a summary of intercorrelations, means, and standard deviations of key study variables for poultry farmers. Similar to that of the beef farmer sample, all key

**Table 2. Intercorrelations, means, Standard Deviations (SD) and Cronbach's αs of key study variables for poultry farmers.**

| Variables | 1. | 2. | 3. | 4. | 5. | 6. | 7. | 8. | 9. | 10. | 11. | 12. | 13. | 14. |
|---|---|---|---|---|---|---|---|---|---|---|---|---|---|---|
| 1. Attitudes | - | .45*** | .51*** | .23*** | .29*** | .08 | .11* | .02 | .04 | .39*** | .29*** | .44*** | .19*** | .11* |
| 2. Subjective norms | | - | .60*** | .30*** | .31*** | - < .01 | .17** | -.01 | .04 | .40*** | .38*** | .41*** | .10* | .06 |
| 3. Perceived behavioral control | | | - | .38*** | .36*** | .08 | .14** | .03 | -.10* | .53*** | .37*** | .50*** | .11* | .07 |
| 4. Openness[+] | | | | - | .35*** | .16** | .22*** | .11* | -.07 | .43*** | .32*** | .29*** | .10* | .07 |
| 5. Conscientiousness[+] | | | | | - | .04 | .11* | -.01 | -.11* | .40*** | .27*** | .29*** | .10* | .06 |
| 6. Extraversion[+] | | | | | | - | .18*** | .06 | .16** | .09 | .04 | .20*** | .03 | .11** |
| 7. Agreeableness[+] | | | | | | | - | .07 | .03 | .22*** | .02 | .23*** | -.02 | -.12* |
| 8. Neuroticism[+] | | | | | | | | - | .13** | -.10* | -.02 | -.03 | .11* | -.14** |
| 9. Present time | | | | | | | | | - | -.12* | -.01 | .05 | .09 | .04 |
| 10. Future time | | | | | | | | | | - | .42*** | .49*** | .18*** | .07 |
| 11. Expected benefits | | | | | | | | | | | - | .36*** | .25*** | .08 |
| 12. Self-efficacy | | | | | | | | | | | | - | .13** | .13** |
| 13. Personal capability concerns | | | | | | | | | | | | | - | .43*** |
| 14. Farm-related concern | | | | | | | | | | | | | | - |
| Mean | 3.91 | 4.15 | 4.15 | 4.28 | 4.32 | 3.34 | 3.82 | 3.41 | 2.79 | 3.83 | 3.98 | 3.84 | 2.93 | 2.87 |
| SD | .62 | .60 | .69 | .90 | .93 | 1.37 | 1.14 | 1.30 | .68 | .49 | .66 | .53 | .69 | .64 |
| Cronbach's α | .57 | .69 | .60 | - | - | - | - | - | .64 | .75 | .82 | .81 | .84 | .83 |

Note.

$^*p < .05$,

$^{**}p < .01$,

$^{***}p < .001$.

[+] 1-item measure, therefore, no Cronbach's α was computed.

correlations were in the expected direction for the poultry farmer sample. Specifically, individuals who were future-oriented also reported positive attitudes, subjective norms, perceived behavioral control, were positive minded (openness to experiences, conscientiousness and extraversion), and expected benefits of engaging in farming. However, there were no significant associations between attitudes and extraversion, neuroticism, and present time; between subjective norms and extraversion, neuroticism, present time orientation, and farm-related concerns; between perceived behavioral control and extraversion, neuroticism, and farm-related concerns; between openness to experience and present time orientation, and farm-related concerns; between conscientiousness and extraversion, neuroticism, and farm-related concerns; between extraversion and neuroticism, future time orientation, expected benefits, and personal capability concerns; between agreeableness and neuroticism, present time orientation, expected benefits, and personal capability concerns; between neuroticism and expected benefits, and self-efficacy; between present time orientation and expected benefits, self-efficacy, and personal capability and farm-related concerns; between future time orientation, expected benefits, and self-efficacy and farm-related concerns, respectively.

## Latent profile analyses

Two separate LPAs were conducted to identify profiles for the beef and poultry farmers, respectively, based on combinations of the 14 profiling variables: attitudes, norms, perceived behavioral control, Big 5 personality traits (openness to experience, conscientiousness, extraversion, agreeableness, and neuroticism), present time orientation, future time orientation, expected benefits, self-efficacy, and farming concerns (personal capability and farm-related

**Table 3. Model fit indices for 2- through 4-profile solutions for beef farmers.**

| Profiles | BIC | *Adj* BIC | VLMR | BLRT | Entropy |
|---|---|---|---|---|---|
| 2 | 18039.57 | 17903.10 | .036 | < .001 | .85 |
| **3** | **17551.88** | **17367.80** | **< .001** | **< .001** | **.92** |
| 4 | 17551.88 | 17367.80 | .103 | < .001 | .85 |

*Note.* N = 471. BIC = Bayesian Information Criteria, VLMR = Vuong-Lo-Mendel-Rubin adjusted test, BLRT = Bootstrapped Likelihood Ratio test. A combination of lowest BIC and adjusted BIC with highest number of profiles and significant *p* values for VLMR and BLRT indicate best fit. Entropy values close to 1 indicate best fit.

concerns). A summary of various model fit indices for 2- through 4-profile solutions is provided in Table 3 for beef farmers.

Results revealed that the 3-profile solution met the criteria for all the relevant fit indices for cattle farmers. In addition to the statistical adequacy, our preferred profile solution also demonstrated practical meaningfulness of the profiles. Therefore, we interpreted the 3-profile solution in the present study. Fig 1 shows the standardized mean scores of the profiling psychological variables for cattle farmers.

Profile 1 (*n* = 67, 14.2% of the sample), labelled as "*Fatalists*", comprised individuals who reported below average scores on attitudes, norms, perceived behavioral control, openness to experience, conscientiousness, agreeableness, extraversion, future time orientation, expected benefits, efficacy to engage in the farming business, and they reported personal capability and farm-related concerns. These farmers also reported average scores on neuroticism, and just above the mean for present time orientation. That is, overall farmers in this segment exhibited a pessimistic psychological mindset with a "living in the present" approach to life. Profile 2 (*n* = 329, 69.9%), labelled as "*Uncommitted or Traditionalists*", was the largest group in the study and comprised individuals who reported average standardized scores on positive attitudes, norms, perceived behavioral control, openness to experience, conscientiousness, extraversion, agreeableness, present- and future-time orientation, expected benefits and efficacy. This group of farmers also reported below average scores on personal capability and farmer-related concerns.

Finally, respondents in Profile 3 (*n* = 75, 15.9%), labelled as "*Positive-minded or Entrepreneurs*", reported above average scores on positive attitudes, norms, perceived behavioral control, conscientiousness, extraversion, neuroticism, present- and future-time orientation,

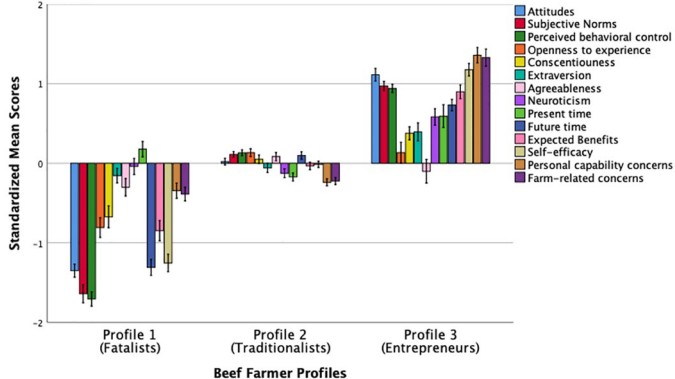

**Fig 1. Standardized mean scores (*M* = 0, *SD* = 1) of different psychological characteristics across three beef farmer profiles.** Error bars represent standard errors (SE) ±1.

**Table 4. Means, standard errors and mean differences across three beef farmer profiles.**

| Variables | *Profile 1* **Fatalists** | *Profile 2* **Tradionalists** | *Profile 3* **Entrepreneurs** | Univariate | |
|---|---|---|---|---|---|
| | (*n* = 67) | (*n* = 329) | (*n* = 75) | *F* (2, 468) | Partial η² |
| Attitudes | 2.86ᵃ (.06) | 3.81ᵇ (.03) | 4.56ᶜ (.06) | 197.54*** | .46 |
| Subjective norms | 2.86ᵃ (.06) | 4.16ᵇ (.03) | 4.80ᶜ (.06) | 277.66*** | .54 |
| Perceived behavioral control | 2.78ᵃ (.06) | 4.19ᵇ (.03) | 4.81ᶜ (.06) | 308.35*** | .57 |
| Openness | 3.25ᵃ (.11) | 4.17ᵇ (.05) | 4.17ᵇ (.11) | 28.53*** | .11 |
| Conscientiousness | 3.52ᵃ (.11) | 4.22ᵇ (.05) | 4.53ᶜ (.11) | 22.94*** | .09 |
| Extraversion | 3.18ᵃ (.15) | 3.30ᵇ (.07) | 3.84ᶜ (.14) | 7.41** | .03 |
| Agreeableness | 3.42ᵃ (.14) | 3.87ᵇ (.06) | 3.65ᵇ (.13) | 4.66* | .02 |
| Neuroticism | 3.13ᵃ (.14) | 3.03ᵃ (.07) | 3.89ᵇ (.14) | 16.37*** | .07 |
| Present time | 2.98ᵃ (.09) | 2.71ᵇ (.04) | 3.30ᶜ (.09) | 20.66*** | .08 |
| Future time | 3.08ᵃ (.05) | 3.79ᵇ (.02) | 4.10ᶜ (.05) | 118.87*** | .34 |
| Expected benefits | 3.19ᵃ (.09) | 3.84ᵇ (.04) | 4.58ᶜ (.08) | 70.81*** | .23 |
| Self-efficacy | 2.94ᵃ (.06) | 3.73ᵇ (.03) | 4.48ᶜ (.06) | 187.42*** | .45 |
| Personal capability concerns | 2.78ᵃ (.08) | 2.87ᵃ (.04) | 4.15ᵇ (.08) | 127.08*** | .35 |
| Farm-related concerns | 2.93ᵃ (.08) | 3.05ᵃ (.03) | 4.23ᵇ (.07) | 119.44*** | .34 |

Note. Means in rows with different superscripts are significantly different at p < .05.

*p < .05,

**p < .01,

***p < .001.

expected benefits and efficacy to engage in the farming business. These farmers also reported above average levels of personal capability and farm-related concerns, with average levels of openness to experience and agreeableness.

One-way MANOVA found significant differences in the psychological variables across the three beef farmer profiles, $F$ (28, 912) = 38.05, $p < .001$; Pillai's Trace = 1.08; partial η² = .54, a large effect size. Post-hoc comparisons, summarized in Table 4, revealed that individuals in Profile 1 reported significantly lower scores on all psychological characteristics than that of Profiles 2 and 3 except for conscientiousness, neuroticism, and personal capability concerns where there were no statistically significant differences between Profile 1 (Fatalists) and Profile 2 (Traditionalists). In contrast, Profile 3 (Entrepreneurs) reported significantly higher scores on all psychological characteristics relative to Profiles 1 and 2 except for openness and agreeableness where there were no significant differences between Profile (Traditionalists) and Profile 3 (Entrepreneurs).

**Table 5. Model fit indices for 2- through 5-profile solutions for poultry farmers.**

| Profiles | BIC | Adj BIC | VLMR | BLRT | Entropy |
|---|---|---|---|---|---|
| 2 | 16465.28 | 16328.83 | < .001 | < .001 | .80 |
| **3** | **16234.72** | **16050.66** | **.06** | **< .001** | **.87** |
| 4 | 16177.68 | 15946.82 | .02 | < .001 | .89 |
| 5 | 16157.07 | 15877.81 | .262 | < .001 | .86 |

*Note. N* = 426. BIC = Bayesian Information Criteria, VLMR = Vuong-Lo-Mendel-Rubin adjusted test, BLRT = Bootstrapped Likelihood Ratio test. A combination of lowest BIC and adjusted BIC with highest number of profiles and significant *p* values for VLMR and BLRT indicate best fit. Entropy values close to 1 indicate best fit.

Table 5 displays various model fit indices for 1- through 5-profile solutions for poultry farmers. Results revealed that the 2- and 4-profile solutions demonstrated all the relevant fit indices in the right direction. However, for the 4-profile solution, the BIC value did not drop substantially from the 3-profile solution. Further, the additional profile group in the 4-profile solution comprised only 19 respondents, making the group sizes unequal for further analyses. On further scrutiny, the 3-profile solution was found to be an improvement from the 2-profile solution as the former resulted in lower BIC and Adj BIC values and greater entropy (.87), even though one of the fit indices (VLMR) did not approach the required significance level (*p* = .06). The 3-profile solution also provided alignment with the similar profile solution obtained for beef farmers, and provided validation of three distinct subgroupings of farmers. Therefore, the 3-profile solution was retained and interpreted in the present study as it provided practical meaningfulness of the poultry farmer profiles obtained.

Fig 2 shows the standardized mean scores of the psychological profiling variables for poultry farmers.

Similar to the beef farmer profile-solution, Profile 1 of poultry farmers (*n* = 26, 6.1% of the sample), labelled as "*Fatalists*", comprised individuals who reported below average scores on attitudes, norms, perceived behavioral control, openness to experience, conscientiousness, agreeableness, extraversion, future time orientation, expected benefits and efficacy to engage in the farming business. These farmers also showed average scores on neuroticism, present time orientation and personal capability concerns; and reported just above the mean for farm-related concerns. Farmers in this segment reported to be concerned about farm-related barriers, however they exhibited a pessimistic psychological mindset to do something about their

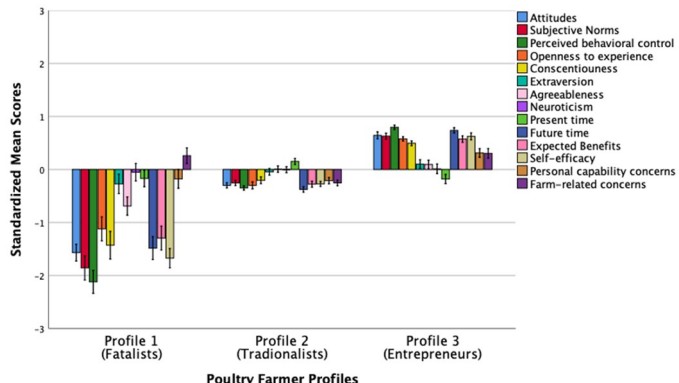

**Fig 2. Standardized mean scores (*M* = 0, *SD* = 1) of different psychological characteristics across three poultry farmer profiles.** Error bars represent standard errors (SE) ±1.

circumstances. Profile 2 ($n$ = 230, 54%), labelled as "*Uncommitted* or *Traditionalists*", was the largest group in the study comprising individuals who reported an overall average standardized scores on all 14 psychological characteristics, i.e., this group of farmers were seen as fence-sitters, just complacent with their current situation. Finally, farmers in Profile 3 ($n$ = 170, 39.9%), labelled as "*Positive-minded* or *Entrepreneurs*", reported above average standardized scores on positive attitudes, norms, perceived behavioral control towards engaging with the farming business, exhibited above average positive personality traits such as openness to experience and conscientiousness, they were future oriented, perceived greater benefits of, and self-efficacy in, managing farm business performance. They also reported having personal capability and farm-related concerns. This group of farmers showed average levels of extraversion, agreeableness, neuroticism, and were just below average on present time orientation.

A second one-way MANOVA found significant differences in the psychological variables across the three poultry farmer profiles, $F(28, 822)$ = 24.82, $p$ < .001; Pillai's Trace = .92; partial $\eta^2$ = .99, a large effect size. Post-hoc comparisons, summarized in Table 6, revealed that individuals in Profile 1 reported significantly lower scores on all psychological characteristics

**Table 6. Means, standard errors and mean differences across three poultry farmer profiles.**

| Variables | Profile 1 Fatalists | Profile 2 Tradionalists | Profile 3 Entrepreneurs | Univariate | |
|---|---|---|---|---|---|
| | (n = 26) | (n = 230) | (n = 170) | F (2, 423) | Partial $\eta^2$ |
| Attitudes | 2.95[a] (.10) | 3.72[b] (.03) | 4.30[c] (.04) | 121.56*** | .37 |
| Subjective norms | 3.04[a] (.09) | 3.99[b] (.03) | 4.52[c] (.04) | 143.69*** | .41 |
| Perceived behavioral control | 2.69[a] (.09) | 3.91[b] (.03) | 4.70[c] (.03) | 311.03*** | .60 |
| Openness | 3.27[a] (.15) | 4.01[b] (.05) | 4.80[c] (.06) | 73.82*** | .26 |
| Conscientiousness | 3.00[a] (.16) | 4.14[b] (.05) | 4.78[c] (.06) | 68.66*** | .25 |
| Extraversion | 2.96[a] (.27) | 3.27[a] (.09) | 3.48[a] (.11) | 2.10 | .01 |
| Agreeableness | 3.04[a] (.22) | 3.83[b] (.07) | 3.93[b] (.09) | 4.66* | .02 |
| Neuroticism | 3.35[a] (.26) | 3.41[a] (.09) | 3.43[a] (.10) | 7.15* | .03 |
| Present time | 2.67[a] (.13) | 2.89[a] (.05) | 2.67[a] (.05) | .05 | < .01 |
| Future time | 3.10[a] (.07) | 3.64[b] (.02) | 4.19[c] (.03) | 159.00** | .43 |
| Expected benefits | 3.19[a] (.09) | 3.84[b] (.04) | 4.58[c] (.08) | 80.30*** | .28 |
| Self-efficacy | 2.95[a] (.08) | 3.69[b] (.03) | 4.17[c] (.03) | 122.81*** | .37 |
| Personal capability concerns | 2.80[ab] (.13) | 2.78[a] (.04) | 3.14[b] (.05) | 14.63*** | .07 |
| Farm-related concerns | 3.04[a] (.12) | 2.71[b] (.04) | 3.06[ac] (.05) | 17.36*** | .08 |

Note. Means in rows with different superscripts are significantly different at p < .05.

*p < .05,

**p < .01,

***p < .001.

than that of Profiles 2 and 3 except for extraversion, neuroticism, and present time where there were no statistically significant differences between the three profiles. In contrast, Profile 3 (Entrepreneurs) reported significantly higher scores on all psychological characteristics relative to Profiles 1 and 2 with an exception for agreeableness where there was no significant difference between Profile 2 (Traditionalists) and Profile 3 (Entrepreneurs). Interestingly, there were also no statistically significant differences in personal capability and farm-related concerns between Profile 1 and Profile 3.

## Discussion

Our study extended previous research into psychological variables implicated in farm business success. We used a profile-based analysis as a means of establishing combinations of a range of psychological characteristics, which formed distinct profiles of enablers of, and barriers to, engaging in smallholder farming. As such, this presents a novel approach within the literature and highlights the utility of looking beyond variable-level associations. The main findings and implications are discussed below.

LPA revealed three qualitatively different profile typologies of both beef and poultry farmers. The first profile arguably captures more fatalist characteristics of both beef and poultry farmers comprising Profile 1 exhibited low scores (below the mean) on all psychological characteristics, with an exception of present time orientation, where this segment showed scores just above the mean. This profile appears to have a closed-minded and pessimistic approach to engaging in farm business. The second profile of beef and poultry farmers was labelled "Uncommitted or Traditionalists" as respondents in this segment exhibited a psychological make-up of those without a strong commitment to engage in an action or do something about improving the current circumstances, being more like fence-sitters. The scores of this sub-grouping hovered around the mean. The third profile, labelled "Positive-minded or Entrepreneurs" comprised individuals who scored above the mean scores for attitudes, subjective norms, perceived behavioral control, conscientiousness, extraversion, neuroticism, present- and future-time orientation, expected benefits and efficacy to engage in the farming business. These farmers reported experiencing personal capability and farm-related concerns, with average levels of openness to experience and agreeableness.

Overall, our findings are consistent with previous research [4] suggesting that positive psychological characteristics such as attitudes, subjective norms, perceived behavioral control, being open to new experiences or opportunities, being conscientious, being future-oriented, expecting benefits of and self-efficacy in engaging in farming are precursors to be progressive and being successful in the business enterprise. This psychological make-up of the farmer profiles is evident in our findings suggesting that Profile 1 respondents reported significantly lower scores on most psychological characteristics compared with other two profile segments compared with Profiles 2 and 3, with the latter profile preforming significantly better relative to other profile segments. We also found that farmers in Profile 3 (Entrepreneurs) also reported significantly greater personal capability and farm-related concerns compared to two other profiles. This makes sense as these farmers are aware of what is going on at their farms and what factors influence their farm business performance, and are proactive in finding solutions to the barriers experience. Subsequent research to the present study [40] explored the associations between profile membership and business performance to assess the predictive validity of psychological profiles of smallholder beef and poultry farmers obtained in the present study.

The next phase of our program of research focused on designing, implementing and evaluating the effectiveness of a range of behavioral-science informed intervention strategies to

improve farm business performance for those farmer profiles who are not optimally performing. Subsequent papers will describe the design of new behavioral intervention strategies and the effectiveness of those interventions in improving the business performance of smallholder poultry farmers. These intervention strategies were customized to best target the preferred learning styles of groups of farmers with similar behavioral profiles to improve their farm business performance, with the experimental treatments ongoing at time of publication of this study.

Only poultry farmers were included in the subsequent intervention study due to the significant time lags required to collect performance data from beef farmers (who market small numbers of cattle once or twice per annum) relative to poultry farmers (who market their products on a daily, weekly or monthly basis) thereby allowing evaluations of the intervention strategies in poultry farmers in months rather than years.

## Limitations and future research directions

As with all research, the present study is not without its limitations. First, the present study used self-report measures, which are susceptible to social desirability. Second, the use of a cross-sectional design limits us making any causal inferences. We also acknowledge that the profiles identified in this study might not reflect existing subgroupings within the actual population [37]. To address this, future studies could employ longitudinal designs to track psychological profile trajectories over time. Third, we found that some of the items comprising the measure (BFI-10) used to assess personality traits were not suitable in the current cultural context. It seems that smallholder farmer respondents were not able to effectively interpret and understand all negatively-worded items. Future research could validate a measure capturing Big 5 personality characteristics taking into account the indigenous cultural backgrounds in South Africa.

## Conclusion

Overall, our findings contribute a novel approach to understanding farmer profiles based on psychological characteristics in a South African beef and poultry smallholder farmer context. We used a profile-based approach to understand the full extent of psychological make-up of different farmer segments capturing the multi-dimensionality of an individual's trait combinations to understand specific facilitators of, and barriers to, engaging in the farm business. In the present study, we identified three qualitatively different profile typologies (*Fatalists*, *Traditionalists*, and *Entrepreneurs*) of beef and poultry smallholder farmers, respectively. Future research could examine the generalizability of farmer profiles obtained in smallholder beef and poultry farmers in South Africa in other African countries and also the association of profile membership with farm business performance.

## Acknowledgments

The authors gratefully acknowledge the inputs of the many research technicians and extension officers responsible for survey data collection amongst smallholder beef and poultry farmers in seven provinces in South Africa.

## Author Contributions

**Conceptualization:** Navjot Bhullar.

**Data curation:** Navjot Bhullar, Nkhanedzeni B. Nengovhela, Livhuwani Mudau, Isaac Koomson.

**Formal analysis:** Navjot Bhullar.

**Funding acquisition:** Nkhanedzeni B. Nengovhela, Heather M. Burrow.

**Investigation:** Navjot Bhullar.

**Methodology:** Navjot Bhullar, Nkhanedzeni B. Nengovhela.

**Project administration:** Nkhanedzeni B. Nengovhela, Livhuwani Mudau, Heather M. Burrow.

**Resources:** Nkhanedzeni B. Nengovhela, Heather M. Burrow.

**Supervision:** Navjot Bhullar, Nkhanedzeni B. Nengovhela, Heather M. Burrow.

**Validation:** Navjot Bhullar.

**Visualization:** Navjot Bhullar.

**Writing – original draft:** Navjot Bhullar, Heather M. Burrow.

**Writing – review & editing:** Navjot Bhullar, Nkhanedzeni B. Nengovhela, Livhuwani Mudau, Renato A. Villano, Isaac Koomson, Heather M. Burrow.

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
