## [Decision Letter · Decision Letter 0]

4 Jan 2022

PONE-D-21-26724Psychological Profiles of South African Smallholder Beef and Poultry FarmersPLOS ONE

Dear Dr. Bhullar,

Thank you for submitting your manuscript to PLOS ONE. After careful consideration, we feel that it has merit but does not fully meet PLOS ONE’s publication criteria as it currently stands. Therefore, we invite you to submit a revised version of the manuscript that addresses the points raised during the review process.

We look forward to receiving your revised manuscript.

Kind regards,

László Vasa, PhD

Academic Editor

PLOS ONE

Journal Requirements:

2. We note that you have referenced (Bhullar, N. (2021)) which has currently not yet been accepted for publication. Please remove this from your References and amend this to state in the body of your manuscript: (Bhullar, N. (2021). [Unpublished]”) as detailed online in our guide for authors http://journals.plos.org/plosone/s/submission-guidelines#loc-reference-style.

“Research undertaken with beef farmers in this study was funded by the Australian Centre for International Agricultural Research [grant number LS/2016/276]. Data derived from poultry farmers in this study was sourced through South Africa’s Poultry Improvement and Recording Scheme, administered by the Agricultural Research Council and funded by the South African Department of Agriculture, Land and Rural Development.”

“Initials of the authors who received each award:

HMB for the ACIAR project and NBN for the DALRRD funds

Grant numbers awarded to each author:

ACIAR LPS/2016/276 and no grant number for the DALRRD funds

The full name of each funder:

Australian Centre for International Agricultural Research; South African Department of Agriculture, Land Reform and Rural Development

URL of each funder website:

https://www.aciar.gov.au/  
https://www.dalrrd.gov.za/ 
https://www.arc.agric.za/arc-api/Pages/Animal-Recording-and-Improvement.aspx (ARC is the administrator of DALRRD funds for the poultry recording scheme)”

Reviewers' comments:

Reviewer's Responses to Questions

**Comments to the Author**

1. Is the manuscript technically sound, and do the data support the conclusions?

Reviewer #1: Yes

Reviewer #2: Partly

2. Has the statistical analysis been performed appropriately and rigorously? 

Reviewer #1: Yes

Reviewer #2: Yes

3. Have the authors made all data underlying the findings in their manuscript fully available?

Reviewer #1: Yes

Reviewer #2: Yes

4. Is the manuscript presented in an intelligible fashion and written in standard English?

Reviewer #1: Yes

Reviewer #2: Yes

5. Review Comments to the Author

Reviewer #1: The topic addressed in the paper is of high importance from the point of view of the South African economy -as indicated in the paper; however, the research question, and the findings might be generalisable or referrred to by other countries as well.

The paper clearly describes the current situation, provides necesary information regarding the research sample and the results and introduces the reader to a detailed analysis of the data. The conclusions are in line with the findings presented.

The paper addresses plenty of international literature, some of which are rather old. Nonetheless, there are some (5) papers from the past five years that underline the topicality of the paper.

I would recommend the authors to search for some fresh literature to replace the old ones, especially regarding the applied methodology.

Reviewer #2: The paper focuses a really challenging and actual topic. Identifying the barriers of engaging agricultural production is essential information for both the academic community and decision makers. The psychological approach is among the most important key factors indeed.

The title of the paper is sound and focusing appropriately, covering and reflecting the content. the The abstract is well written and well understandable. Keywords are well selected.

The introduction is acceptable, highlighting the importance of the topic and the context in a proper way.

I am not satisfied with the literature review. Some essential part of the sources are overviewed but in fact, only those which were used to determine the methodology an setting the context. However, the "real" literature overview, where the works of other authors who are (were) dealing with the same topic is missing. I recommend to set a critical, analytical and comparative literature review after the introduction part.

The methodology description is well written, the results are supported by the methods.

The conclusion part is unacceptable short, the authors should extend it, based on the results.

6. PLOS authors have the option to publish the peer review history of their article (what does this mean?). If published, this will include your full peer review and any attached files.

Reviewer #1: **Yes: **Prof. Dr. Kornélia Lazányi

Reviewer #2: No

---

## [Author Response · Author response to Decision Letter 0]

28 Feb 2022

Information provided here is similar what is in the Response Letter that is attached. 

Reviewer #1: 

1. The topic addressed in the paper is of high importance from the point of view of the South African economy - as indicated in the paper; however, the research question, and the findings might be generalisable or referred to by other countries as well.

Response: Thank you for your suggestion. However, given the characteristics of the sample of smallholder farmers in South Africa, we are cautious in making any such claim about the generalisability of our findings. We have now noted that our findings could be replicated in different contexts and other countries in the African continent at p. 29-30. 

2. The paper clearly describes the current situation, provides necessary information regarding the research sample and the results and introduces the reader to a detailed analysis of the data. 

Response: Thank you. 

3. The conclusions are in line with the findings presented.

Response: Thank you.

4. The paper addresses plenty of international literature, some of which are rather old. Nonetheless, there are some (5) papers from the past five years that underline the topicality of the paper. I would recommend the authors to search for some fresh literature to replace the old ones, especially regarding the applied methodology.

Response: We have now included further relevant research literature pertaining to key psychological predictors of farm business performance at p. 5. 

Reviewer #2: 

1. The paper focuses a really challenging and actual topic. Identifying the barriers of engaging agricultural production is essential information for both the academic community and decision makers. The psychological approach is among the most important key factors indeed.

Response: Thank you for the positive feedback.

2. The title of the paper is sound and focusing appropriately, covering and reflecting the content. The abstract is well written and well understandable. Keywords are well selected.

Response: Thank you.

3. The introduction is acceptable, highlighting the importance of the topic and the context in a proper way.

Response: Thank you.

4. I am not satisfied with the literature review. Some essential part of the sources are overviewed but in fact, only those which were used to determine the methodology an setting the context. However, the "real" literature overview, where the works of other authors who are (were) dealing with the same topic is missing. I recommend to set a critical, analytical and comparative literature review after the introduction part.

Response: As suggested, we have now revised the literature review section to provide an overview of literature examining an association between key psychological variables and farm business performance (see p. 5). 

5. The methodology description is well written, the results are supported by the methods.

Response: Thank you.

The conclusion part is unacceptable short, the authors should extend it, based on the results.

Response: As suggested, we have now included additional information at p. 29-30. 

Journal Requirements:

Response:

2. We note that you have referenced (Bhullar, N. (2021)) which has currently not yet been accepted for publication. Please remove this from your References and amend this to state in the body of your manuscript: (Bhullar, N. (2021). [Unpublished]”) as detailed online in our guide for authors http://journals.plos.org/plosone/s/submission-guidelines#loc-reference-style.

Response: Thank you. We have now revised the manuscript text. 

“Research undertaken with beef farmers in this study was funded by the Australian Centre for International Agricultural Research [grant number LS/2016/276]. Data derived from poultry farmers in this study was sourced through South Africa’s Poultry Improvement and Recording Scheme, administered by the Agricultural Research Council and funded by the South African Department of Agriculture, Land and Rural Development.”

“Initials of the authors who received each award:

HMB for the ACIAR project and NBN for the DALRRD funds

Grant numbers awarded to each author:

ACIAR LPS/2016/276 and no grant number for the DALRRD funds

The full name of each funder:

Australian Centre for International Agricultural Research; South African Department of Agriculture, Land Reform and Rural Development

URL of each funder website:

https://www.aciar.gov.au/
https://www.dalrrd.gov.za/
https://www.arc.agric.za/arc-api/Pages/Animal-Recording-and-Improvement.aspx (ARC is the administrator of DALRRD funds for the poultry recording scheme)”

Response: Thank you. We have removed funding information provided in the manuscript. 

Response: Thank you, we have checked the Reference list for accuracy.

---

## [Editor Report · Decision Letter 1]

7 Mar 2022

Psychological Profiles of South African Smallholder Farmers

PONE-D-21-26724R1

Dear Dr. Bhullar,

We’re pleased to inform you that your manuscript has been judged scientifically suitable for publication and will be formally accepted for publication once it meets all outstanding technical requirements.

Kind regards,

László Vasa, PhD

Academic Editor

PLOS ONE
---

## [Editor Report · Acceptance letter]

6 Jan 2023

PONE-D-21-26724R1 

Psychological Profiles of South African Smallholder Farmers 

Dear Dr. Bhullar:

I'm pleased to inform you that your manuscript has been deemed suitable for publication in PLOS ONE. Congratulations! Your manuscript is now with our production department. 

Kind regards, 

on behalf of

Prof. Dr. László Vasa 

Academic Editor

PLOS ONE